# Attention Deficit Hyperactivity Disorder and Bipolar Disorder: Diagnosis, Treatments, and Clinical Considerations: A Narrative Review

Amber N. Edinoff [1,*], Tucker L. Apgar [2], Jasmine J. Rogers [3], Joshua D. Harper [3], Elyse M. Cornett [4], Adam M. Kaye [5] and Alan D. Kaye [4]

1    Department of Psychiatry and Behavioral Medicine, Louisiana State University Health Science Center Shreveport, Shreveport, LA 71103, USA
2    Department of Chemical Biology and Biochemistry, Vanderbilt University, Nashville, TN 37235, USA; tuckerapgar22@gmail.com
3    School of Medicine, Louisiana State University Health Shreveport, Shreveport, LA 71103, USA; JJR001@lsuhs.edu (J.J.R.); JDH001@lsuhs.edu (J.D.H.)
4    Department of Anesthesiology, Louisiana State University Shreveport, Shreveport, LA 71103, USA; elyse.bradley@lsuhs.edu (E.M.C.); alan.kaye@lsuhs.edu (A.D.K.)
5    Department of Pharmacy Practice, Thomas J. Long School of Pharmacy and Health Sciences, University of the Pacific, Stockton, CA 95211, USA; akaye@pacific.edu
*    Correspondence: amber.edinoff@lsuhs.edu

**Abstract:** Attention-deficit Hyperactivity Disorder is one of the most common childhood mental health disorders, affecting about 5.6% of the population worldwide. Several studies have specifically shown a high prevalence of comorbid mood disorders, such as depression and bipolar disorder (BD), in those diagnosed with ADHD. Several common symptoms of ADHD are also found in BD, which are characterized by alternating periods of euthymia and mood disturbances. The inattention and impulsivity of ADHD can be seen in manic and hypomanic episodes of BD. Over the past decade, there has been an increased interest in research between the correlation of ADHD and pediatric bipolar disorder (PBD) in children. Some experts hypothesize that more children are comorbidly diagnosed with ADHD and PBD because of how many clinicians treat children with ADHD. Other factors, which may affect the dual diagnoses of ADHD and PBD, are overlapping diagnostic criteria for the two disorders, the inevitable biases seen when one disorder is diagnosed without the other, and related risk factors leading to prodromal relationships. By examining clinical trials, a better understanding of whether ADHD and PBD have a stepwise progression or if other factors influence these comorbidities, such as blurred lines of diagnostic criteria. Those with ADHD are also at an increased risk of impairment at work and in social settings. This manuscript explores both progression of this disease and its clinical connections to other disorders.

**Keywords:** ADHD; bipolar disorder; pediatrics; stimulants; comorbidity





## 1. Introduction

Attention-deficit Hyperactivity Disorder is one of the most common childhood mental health disorders, affecting about 5.6% of the population worldwide, without respect to socioeconomic factors [1]. It is characterized by significant motor hyperactivity and inattentiveness that can affect social functioning and cause impairment in other settings [2]. Symptoms result from dysfunction in the frontal cortex and basal ganglia, with treatment to regulate neurotransmission in these areas [3]. This disorder is usually clinically diagnosed through current and prior symptoms and impairment [1,4]. It is now established that symptoms can persist into adulthood in up to 65% of those diagnosed [3]. Children diagnosed with attention-deficit hyperactivity disorder (ADHD) are more likely to experience additional mental health issues in adulthood [2]. These disorders include mood disorders, such as bipolar disorder (BD), and personality and anxiety disorders. The prevalence of

adults seeking treatment for at least one additional disorder following an ADHD diagnosis can vary from 57 to 97% [2].

Several studies have specifically shown a high prevalence of comorbid mood disorders, such as depression and BD, in those diagnosed with ADHD [1,2]. In a survey conducted in the US, the National Comorbidity Survey-Replication found the prevalence of comorbid BD in patients diagnosed with ADHD to be 21.2% [2]. Several common symptoms of ADHD are also found in BD, which are characterized by alternating periods of euthymia and mood disturbances. The inattention and impulsivity of ADHD can be seen in BD's manic and hypomanic episodes [2]. Patients with BD can also present from resulting cognitive dysfunction during the premorbid phase into the early course of the disease. Cognitive dysfunction usually results in impairments in learning, memory, and executive function. Comorbid ADHD and treatment with stimulants have been associated with an earlier onset of BD in those with past exposure to stimulants [2]. These comorbid conditions can also complicate the progression and treatment of BD [5].

ADHD and BD are both highly heritable and impairing conditions that have been shown to have several similarities and similar links [4,5]. The effectiveness of pharmacologic treatment strategies has been proven to be very effective. However, the long-term effectiveness and consequences of the medications are not yet well understood [1,4]. One of the major areas of future research on ADHD will be to study the long-term effects, both positive and negative, of the treatment of this disorder [3]. Moreover, the relationship between ADHD and BD and the progression of each condition has been an area of study. There have been longitudinal studies to assess the development of hypomanic episodes in children with ADHD [2,5]. The aim of this manuscript is to discusses the relationship and clinical connections between the two disorders, including the medication induced mania hypothesis, clinical overlap, and genetic disposition in a narrative review format. It also explores effects of treatment on progression of both ADHD and BD.

## 2. Attention Deficit Hyperactivity Disorder Overview

ADHD exhibits a complex etiology integrating genetic and environmental factors for studied cases [6–9]. Further updates in the research literature have advanced our understanding of the disorder [10]. Existing family and twin studies support genetics as an important factor in the onset of ADHD [11]. Specifically, twin studies have concluded that the heritability of ADHD is estimated to be 60–90% [11–14]. However, when evaluated based on isolated genes and risk variants, ADHD demonstrates a poor correlation to any specific gene [15,16].

ADHD typically has had an onset at a young age. In 2013, the DSM V updated the age limit for the onset of qualifying symptoms for ADHD from age 6 to age 12 to prevent the misdiagnosis of normal inattention symptoms in the pediatric population [17]. For diagnosis, adolescents (17 years or older) can display less hyperactive or inattentive symptoms (at least five). Increasingly, it has become accepted that ADHD can exhibit a late manifestation or maintain symptoms throughout adulthood [18–21]. As a baseline for diagnosis, ADHD must also be present in at least two settings where symptoms present in various environments like the home and school.

### 2.1. Pathophysiology

ADHD presents a variety of cognitive and functional deficits which originate from abnormalities in the brain [22]. The two most common theories surrounding the mechanisms of ADHD involve a top-down and bottom-up model. Top-down models emphasize cognitive control and executive functioning. Specifically, abnormal executive function manifesting as impaired response inhibition has been used to explain the cognitive symptoms of ADHD [23–26]. In contrast, bottom-up models emphasize motivational, incentive, or reward responses.

The use of brain imaging has helped researchers elucidate the mechanisms behind ADHD [27,28]. Specifically, structural imaging studies using MRI have determined abnor-

mal symptoms arising from a smaller size of the cerebrum, cerebellum, anterior cingulate cortex, and dorsolateral prefrontal cortex. Imaging studies also support theories about brain developmental patterns, as studies evaluating cortical thickness reported a delay in growth among ADHD patients [29,30]. An imaging study using MRI found widespread changes in the maturation of white matter fiber bundles and gray matter density in the brain which lead to structural shape changes in the middle and superior temporal gyrus, and fronto-basal portions of both frontal lobes [31]. On the circuitry level, dysfunctions in the super longitudinal fasciculus and cortico-limbic areas are found in those with ADHD [31]. These morphological findings predicted an ADHD diagnosis correctly in up to 83% of the cases in this imaging study [31]. Furthermore, imaging for ADHD has also supported top-down and bottom-up theories by visualizing the structural environment of the brain [32].

*2.2. Epidemiology*

Previous studies have estimated that ADHD affects approximately 2–7% of the global population [33–36]. Subsequently, the prevalence of ADHD in adults (between 19 and 45 years) has been estimated at 2.5% [37] where 40–60% exhibit partial remission of symptoms [38]. The most comprehensive meta-analysis, which was conducted by Polanczyk and colleagues, estimated a prevalence of 5.29% based upon 153 evaluated studies [34]. Variability in prevalence values reflected differences in diagnostic criteria, information sources, and functional impairment requirements [33,38,39]. Additionally, point prevalence did not change from 1985 to 2012 or between geographical regions across the globe [34,39].

## 3. Treatment

Pharmacologic treatments for ADHD can be divided into non-stimulants and stimulants. Stimulants, including amphetamine and methylphenidate-based medications, are considered first-line in treating ADHD in children and adults [40]. These stimulants boost arousal in the prefrontal cortex by increasing norepinephrine and dopamine concentrations in the brain [41]. Increasing synaptic dopamine and catecholamine are main mechanisms of action of both methylphenidate and amphetamine. However, there are specific differences that affect how they alter concentrations [40]. The stimulant methylphenidate exerts its effects by inhibiting the dopamine transporters (DAT) and norepinephrine transporters (NET). It increases dopaminergic neurotransmission by inhibiting presynaptic DAT and, as a result, reuptake at the synapse [40,41].

Studies have demonstrated that methylphenidate directly interacts with adrenergic receptors to stimulate excitability in the cortex via activation of a2 adrenergic receptors [40]. Methylphenidate also exerts agonistic activity at the 5-HT1a receptor [40]. This results in the elevation of extracellular dopamine and norepinephrine levels, and as a result, increased binding to their respective receptors [40].

Another stimulant used in treating ADHD is an amphetamine salt, whose primary mechanism of action is to increase catecholamine release [40]. It acts as a competitive inhibitor of DAT and as a pseudo-substrate at norepinephrine binding sites. It acts to increase dopamine release by inhibiting the VMAT-2, or vesicular monoamine transporter 2, and it also inhibits monoamine oxidase activity to decrease the cytosolic breakdown monoamines [41]. Both methylphenidate and amphetamine increase dopamine release, which increases responsiveness to external stimuli [40,41]. These stimulants have *d* and *l* isomers, with the *d* isomer being more potent than its *l* counterpart. Specifically, the *d* isomer is more potent at the norepinephrine transporter and dopamine transporter binding sites b [41].

Non-stimulant medication treatment includes atomoxetine and bupropion. Although the mechanism is not completely clear, atomoxetine is thought to selectivity inhibit norepinephrine uptake and preferentially binds to areas of known high distribution of noradrenergic neurons [42]. Bupropion is an anti-depressant that has a variety of uses, including depression, anxiety, ADHD, and smoking cessation. Its mechanism of action likely involves the reuptake inhibition of the catecholamines, dopamine, and noradrenaline [43]. This

mechanism is like the one for psychostimulants, but bupropion is not a controlled substance. Clonidine is another choice, which is an alpha 2 adrenergic receptor agonist. Clonidine is good for impulsivity and hyperactivity but not useful for symptoms of inattention [44]. Guanfacine is a direct alpha 2α subtype agonist within the central nervous system leading to reduced peripheral sympathetic outflow and strengthening of regulation of both attention and behavior within the prefrontal cortex through modulation at norepinephrine receptors [44]. Its actions are found in the locus coeruleus and can result in improved attention [44]. Clonidine and Guanfacine are considered second or third-line treatments after the use of stimulants has failed [45].

## 4. Attention Deficit Hyperactivity Disorder and Bipolar Disorder

There is an overlap in terms of symptoms when looking at ADHD and BD. These two disorders often co-occur, and they are associated with worse outcomes [46]. There is evidence that children with ADHD have a higher risk of being diagnosed with BD later in life [47]. There have been attempts to develop neuropsychological testing that can better identify those with both disorders. However, research has shown that these tests have limited power to differentiate between BD adults with and without childhood ADHD [46].

Symptoms of both ADHD and BD can have overlap which will be further explored in the next section. Mania is associated with bipolar disorder and is highlighted with increased energy and disorganized thinking with the inability to plan which can be seen in the increased goal-related activity. These are things that are also seen in ADHD. Other mood disorders such as depression and anxiety can have decreased concentration, which can also be seen in ADHD. So, at times, it can be hard to separate what diagnosis is correct because of the overlapping symptoms.

Treatment is a question that has been looked at in the research as the overlapping symptoms that cause issues and distress to the patient may be worsened by traditional ADHD treatment of stimulants. There is concern that the use of stimulants may worsen mania and psychosis and further debilitate the patient [47]. A study performed by Galanter et al. suggests that youth treated with stimulants did not have an increase in mania or psychotic symptoms [48]. Furthermore, children in the first month of treatment who had ADHD symptoms and manic symptoms had a more robust response to stimulant treatment. However, the studies in adults are less promising. In a review of three cases, the use of stimulants increased psychosis in patients [49]. The authors concluded that patients need to be monitored for adverse responses to treatment. In the next section, the review will look at the clinical studies regarding the overlap of ADHD and BD.

*Genome Wide Association Studies*

There have been studies that have looked at genetic loci as conferring genetic risk for both bipolar disorder and ADHD. Demonitis et al. performed a genome-wide association study (GWAS) on 20,183 individuals with ADHD and 35,191 controls [50]. The study found around 2932 genes that showed significant association with ADHD [50]. They found that there was also a genetic correlation with both schizophrenia and bipolar disorder, however, this was not significant.

Another GWAS was performed by Mullins et al. looked at genes that could be associated with the increased risk of developing BD [51]. They performed a GWAS of 41,917 BD cases and 371,549 controls of European ancestry. Their study found 64 associated genomic loci in their study population. These risk alleles were genes in the synaptic signaling pathways and brain-expressed genes, which had high specificity of expression in neurons of the prefrontal cortex and hippocampus [51]. These are also areas that are implicated to be dysfunctional in ADHD; however, the authors made no connections to ADHD in their study so only references can be drawn.

## 5. Methods

Studies regarding the clinical overlap, medication-induced mania theory of BD, and co-occurrence of ADHD and BD were searched on PubMed only. Access to other scholarly search engines was not performed as access was lacking to them by the institution. Only articles written in English were included. Search terms of articles included ADHD and Bipolar Disorder, along with the terms overlap, co-occurrence, and genetics. Articles must have been published within the past 20 years for them to be included in this narrative review. There were no articles that were systematically excluded as this was a narrative review.

## 6. Clinical Studies

Over the past decade, there has been an increased interest in research between the correlation of ADHD and pediatric bipolar disorder (PBD) in children. Clinical trials have been performed and reviewed with multiple hypotheses in mind to attempt to determine if there is a valid association between the two, and if one of the disorders precedes in diagnosis. Some hypothesize that more children are comorbidly diagnosed with ADHD and PBD because of how many clinicians treat children with ADHD. Other factors that may affect the dual diagnoses of ADHD and PBD are overlapping diagnostic criteria for the two disorders, the inevitable biases seen when one disorder is diagnosed without the other, and related risk factors leading to prodromal relationships. By examining clinical trials, a better understanding as to whether ADHD and PBD have a stepwise progression or if other factors influence these comorbidities, such as blurred lines of diagnostic criteria. On the one hand, if the information gathered shows a relationship between these two diagnoses, it will be helpful because precautions can be enacted to prevent the progression from ADHD to BD or vice versa. On the other hand, if the comorbidity of the two disorders is occurring because of a bias or blurred diagnostic criteria, then a necessary change can be made to prevent unnecessary and inaccurate comorbid diagnoses in the pediatric population.

This diagnosis often is accompanied by alternative diagnoses as the child ages. According to a study conducted on a Medicaid population of 22,797 children diagnosed with ADHD in South Carolina, 40.1% were concurrently diagnosed with conduct disorder (CD)/oppositional defiant disorder (ODD), 15% were diagnosed with anxiety disorder, 5.7% were diagnosed with substance use disorder, and 7.8% were diagnosed with BD [52]. Other studies have found the comorbid incidence of PBD and ADHD to fall in the range of 8% to 16.5% [53,54]. The varying percentages of comorbidity is a statistic that should be investigated because the prevalence is above the rates of chance alone. There are currently few explanations for the disparity, such as the size of the population being sampled, surveillance bias, and the cohorts in certain studies from specialized clinics, potentially leading to a form of Berkson's bias. A more precise incidence of comorbid diagnoses would allow for a better understanding of the two disorders' effects on one another in our pediatric population. A better understanding of the correlation between these two disorders is important because it could save lives since a diagnosis of PBD comes with an increase in morbidity and mortality.

### 6.1. False Correlation Hypotheses

Bias can creep its way into research in many different forms and often has the chance to skew scientists' perspectives of various observations. One form of bias that could be presenting a false correlation between ADHD and PBD is surveillance bias. When children are labeled with a diagnosis of ADHD, they are placed in a category of children that are assessed with a bias by examining them for signs and symptoms of other diagnoses. Clinical trials have shown that those diagnosed with ADHD tend to be assigned additional diagnoses later in their life. One of the major comorbidities diagnosed included CD/ODD. With these diagnoses, one would expect to see these patients more often in a clinical setting due to the disruptions in the home and at school that often occur with these disorders. The behaviors of these categorized children can mimic symptoms of PBD, and the repeated visits to see a physician can lead to more of these children being diagnosed with PBD. This

surveillance bias has the potential to skew the number of patients accurately diagnosed with PBD and can lead to greater symptom severity, greater functional impairment, and further comorbid diagnoses [54].

### 6.2. Overlapping Criteria

Besides bias playing a part in the misdiagnosis of PBD, the criteria required for a diagnosis of ADHD or PBD could be an issue of concern. The DSM-5 is continuously under revision, and recently there have been changes to certain aspects of diagnostic criteria for ADHD and BD. When studying these diagnostic criteria between these two disorders, we find there is considerable overlap. Patients with ADHD frequently have a vast amount of energy, causing impairment in their performance at school and in the home. With increased overall energy, children with ADHD can also commonly have psychomotor impairment, restlessness, forgetfulness, fidgeting, distractibility, and impulsivity. All of these symptoms in a child with ADHD tend to look very similar and may be mistaken as a manic episode [55]. Figure 1 below shows a Venn diagram of the overlapping and unique symptoms seen in patients with ADHD and BD, respectively. Related to similar diagnostic criteria between these diagnoses, once ADHD is diagnosed in a child, they immediately meet multiple criteria required to be diagnosed with BD. The same can also be said for patients that are diagnosed with BD first. Suppose the DSM-5 were to be revised to differentiate the criteria of these two disorders a little further. In that case, it could decrease the burden of false diagnoses and morbidity/mortality that accompanies BD.

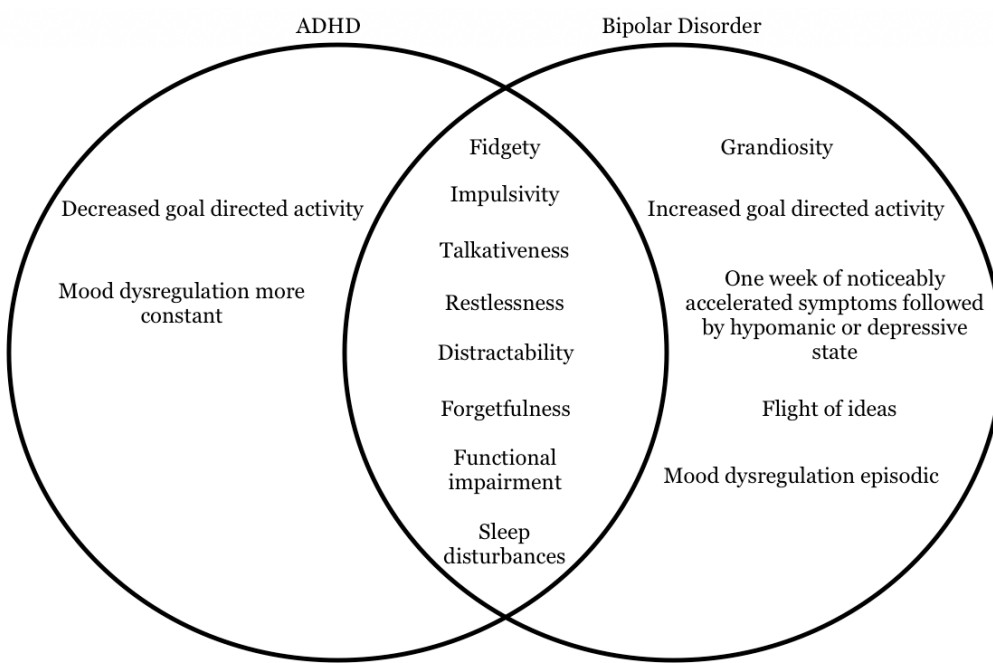

**Figure 1.** Diagnostic criteria of attention deficit hyperactivity disorder versus bipolar disorder.

### 6.3. Correlation of Risk Factors

Before children are diagnosed with ADHD or BD, a thorough exam of the patient's background must be completed. Aside from the symptoms required for diagnoses mentioned previously, other aspects of these patients' lives that are important to consider are family history of mental health problems, a good understanding of home life, and past abuse or trauma. In both ADHD and BD patients, genetics is one of the most important risk factors in determining the probability of future diagnosis with either disorder mentioned. A study on a cohort of children that had no previous diagnosis of ADHD found a significant relationship between the likelihood of being diagnosed with ADHD and parental mental health problems [56]. Another study analyzing probands diagnosed with

BD discovered that the following generations' odds ratio of being diagnosed with BD was equal to seven [57]. These studies have found that certain genes are associated with ADHD and BD, but these studies did not separate the genetic aspect from being raised in the home by an adult dealing with a mental health problem.

Further research should be performed to eliminate potential confounding bias by separating the genetics aspect from the upbringing aspect since they are both major risk factors. For both ADHD and BD, community involvement, family climate, and self-efficacy are strong protective factors in preventing a mental health problem [56]. Similar to how the diagnostic criteria of these two disorders are similar, so are the risk factors and protective factors. The blurred lines of diagnostic criteria can lead to more false-positive diagnoses of BD and ADHD. Still, the risk factors could be a possible explanation for these two disorders coinciding with one another.

### 6.4. Medication-Induced Mania Theory

When ADHD is diagnosed in the United States, there are a couple of options for treatment, as previously mentioned. With mainstay therapy in the US being stimulants, some speculate that the overuse of stimulants could induce manic symptoms in the pediatric population. When these patients are treated with an anti-depressant, without a thorough examination to rule out BD, treatment-induced mania or hypomania is possible. A study examining the frequency of PBD diagnoses in the United States compared to the Netherlands gives good insight into such a correlation. When comparing these two countries, there are stark differences in the number of patients given stimulants and anti-depressants as treatments for ADHD and depression, respectively [58]. The research went on to show no significance of the stimulant treatment for ADHD and induction of BD but did, however, point out that one of the first signs of BD disorder in pediatric patients was depression. Therefore, the increase in PBD patients in the United States compared to the Netherlands could be attributed to anti-depressants.

A study conducted on a large cohort of pediatric patients from Taiwan examined the usage of methylphenidate and atomoxetine in those diagnosed with ADHD. Groups were separated into short-term treatment, long-term treatment, and no treatment and then followed up to see if any manic episodes or symptoms had developed. Their research showed that the odds of being diagnosed with BD while on long-term therapy of methylphenidates were decreased. The odds of being diagnosed with BD while on short-term methylphenidate treatment were not significant compared to the ADHD group that was not receiving treatment. When it came to atomoxetine, the odds of being diagnosed with BD for long-term and short-term treatment patients were not connected [59]. One significant aspect discovered during this study was when compared to the control group of pediatric patients, those diagnosed with ADHD were seven times more likely to be diagnosed with BD at some point in their lives. This study shows some good data on the short-term effects of stimulants and the odds of being diagnosed with BD. When it comes to long-term odds of showing BD symptoms, it was capped at 12 years. This inevitably could cause researchers to miss some emergent cases of BD.

Another study performed examined BD patients that were taking methylphenidates with and without mood stabilizers. With the high percentage of BD patients also having comorbid ADHD, it is important to see if methylphenidates treating the ADHD symptoms also affects these patients' BD eliciting more manic episodes. This study showed that the risk of developing mania while using methylphenidate as a monotherapy elevated the number of incidences, but it was not statistically significant. There was no increase in the risk of treatment-induced mania when the patients took methylphenidates while concomitantly taking a mood stabilizer [60].

Stanford University School of Medicine conducted a study of anti-depressants usage in our pediatric population with depression and stimulants in pediatric ADHD patients. This research was performed with interest in these treatments and their chances of inducing mania. The research concluded, based on multiple sources, that stimulants did not increase

treatment-induced mania and may even be protective against manic episodes. More manic episodes were seen when SSRIs were used as a monotherapy for BD with depression. SSRIs and mood stabilizers, when used together, work well to treat pediatric patients with BD with depressive symptoms. This study also showed that if stimulants become problematic during treatment or ineffective, atomoxetine can be used and is relatively safe [60]. See Table 1 below for a summary of studies mentioned on treatment-induced mania.

**Table 1.** Clinical trials examining the use of medications for ADHD and their likelihood of inducing manic-like behavior in the pediatric population.

| Study | Conclusion | Objective and Population | Method | Results |
|---|---|---|---|---|
| Jerrell et al. [2] 2014 | ADHD and BP co-occurrence has increased. Further investigation of risk due to comorbid conditions and pharmacotherapies is examined as cause. | Retrospective cohort of 22,797 inpatient and outpatient children age 17 and under with prior ADHD diagnosis. Modification diagnoses and medication prescriptions studied. | The adjusted odds ratio (aOR) for a child with ADHD developing BD were significant for CD/ODD (aOR = 4.01), anxiety disorder (2.39), substance use disorder (1.88), methylphenidate or atomoxetine (1.01), fluoxetine (2.00), sertraline (2.29), bupropion (2.22), trazodone (2.15), venlafaxine (2.37). | A diagnosis of BD was more likely in individuals with cluttering specific patterns of comorbid psychiatric disorders and more likely with medications for the treatment of depression. |
| Wang et al. [9] 2015 | Determine if the pharmacotherapy for ADHD influences the risk of developing BD. | Cohort of 144,920 newly diagnosed Taiwanese ADHD patients between 1999 and 2011. Cohort separated into non-users, short-term users, and long-term users of methylphenidates or atomoxetine. | The ADHD group showed a significantly increased risk of developing BD than the control group (2.1% vs. 0.4%, respectively). Long-term users of methylphenidate were less likely to develop BD (aOR = 0.72). All other relationships were found to be insignificant. | Long-term use of methylphenidates was protective, while short-term use was not connected to increased risk of BD. Long-term and short-term use of atomoxetine was not connected to an increased risk of developing BD. |
| Vicktorin et al. [10] 2017 | Risk of treatment-emergent mania associated with methylphenidates as monotherapy and concomitantly with mood stabilizers in BD patients. | Cohort of 2307 Swedish adults with BD initiated therapy with methylphenidates between 2006 and 2014. Separated into a methylphenidate monotherapy group and methylphenidate plus mood stabilizer group. | Within 3 months on monotherapy, manic episodes increased. Patients taking mood stabilizers had a lower risk of mania after starting stimulants. | Careful assessment should be conducted to rule out BD before initiating monotherapy with psychostimulants. |
| Study | Conclusion | Objective and Population | Method | Results |
| Goldsmith et al. [11] 2011 | Examine whether exposure to psychostimulants or antidepressants precipitates or exacerbates manic symptoms or decreases the age at onset of mania in pediatric populations. | Literature review | Risk for mania in ADHD treated with stimulants was relatively low. Children with depression and/or anxiety had a low risk (<2%) of antidepressant-induced mania (AIM). The risk of general activation after SSRI treatment was greater (2–10%). However, AIM was seen to be even higher with children treated in specialized clinics. | For BD and ADHD patients, effective mood stabilization is important before adding a stimulant. There was no clear evidence that stimulants or SSRIs accelerate the natural course of BD development. Prescribers should proceed cautiously when using these agents in youth already at risk for developing BD. |

There is an increased incidence of depression, ADHD, BD, and other serious disorders across the board in our pediatric patient population. With the literature currently published in journals, there does not seem to be an association between ADHD and medication-induced BD. There are overlapping criteria between the two, and it would be advantageous to further differentiate the criteria of these two disorders. More research should be conducted on depression and its correlation with ADHD and PBD. It is known that patients with ADHD have an increased likelihood of having depressive symptoms. These patients have a high chance of being prescribed an SSRI in the United States to treat the depressive symptoms, but if these patients are not screened for BD, it could increase manic episodes and overall diagnoses of BD.

## 7. Limitations of the Manuscript

Limitations of this manuscript include the fact that this is a narrative review. The evidence is reported in a narrative format and the quality of the evidence was not weighed for inclusion or exclusion. The reason for this, with the information gathered being presented in this way, it that it enables the reader to draw conclusions for themselves. Another limitation is that none of the studies conducted above were performed by the authors.

## 8. Conclusions

ADHD is a common mental health disorder that is complex. Symptoms and variation among patients are most likely due to the variation of pathophysiology in the prefrontal cortex. There have been previous links of genetic and environmental factors associated with ADHD. Its progression in those diagnosed is often complicated by comorbid disorders, including substance use disorders, mood disorders, and personality disorders. Those with ADHD are also at an increased risk of impairment at work and in social settings. This manuscript attempts to explore both progression of this disease and its clinical connections to other disorders.

The clinical connections and links between ADHD and BD have recently been investigated and studied through clinical trials. There are currently hypotheses that center around the potential association between comorbid disorders in children. Those diagnosed with ADHD are at increased risk of comorbid mood disorders, such as BD. These disorders also share several symptoms that may make distinguishing the disorders difficult. According to some clinical trials, this link may be partly related to how we treat and manage these disorders. The management has classically involved both pharmacotherapy and psychosocial therapy, with stimulants being the first line of treatment. Specifically, the use and side effects of stimulants have been studied in those diagnosed with ADHD, along with the relationship between SSRI use in ADHD for managing depressive symptoms and the risk of subsequent mania.

In conclusion, there are currently several established links between ADHD and other conditions, such as BD. These links seem to be due, at least in part, to the side effects of the pharmacological treatment. Variation in dopaminergic receptors in the brain may contribute to the different effects of stimulants and is an area for future research in the field.

**Author Contributions:** Conceptualization, A.N.E.; writing—original draft, A.N.E., J.D.H., J.J.R. and T.L.A.; writing—review and editing, A.N.E., E.M.C., A.M.K. and A.D.K. All authors have read and agreed to the published version of the manuscript.

**Funding:** This research received no external funding.

**Data Availability Statement:** The data in the manuscript can be found on the pubmed database of articles.

**Conflicts of Interest:** The authors have no conflict of interest to declare.

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
