# Peer review of "Attention Deficit Hyperactivity Disorder and Bipolar Disorder: Diagnosis, Treatments, and Clinical Considerations: A Narrative Review"

_2673-5318, doi:10.3390/psychiatryint3010002_

Round 1

Reviewer 1 Report

The manuscript by Edinoff et al., is an interesting research review on the potential roles of ADHD and treatment and their links to the mood problems. The authors seem to focus on the bipolar disorders (BD) and discuss the diagnostic aspects. The study design is clinically relevant and the issues in manuscript have been well written. There are some concerns:

  1. The rationale of studying ADHD and juvenile-onset BD correlation is not fully supported by the manuscript.
  2. A higher “prevalence of comorbid mood disorders” has been mentioned (such as Line 16). Mood disorders and manic-like behaviors should be described in the manuscript in details.
  3. As discussed in Lines 95-96, the involved brain regions can be summarized for the study among different related diseases.

Author Response

  1. The rationale of studying ADHD and juvenile-onset BD correlation is not fully supported by the manuscript.

            Answer: Thank you for this point. It’s not really a correlation that we’re trying to draw but a possible difference in the two. There is also not a lot of studies that look at the direct correlation.

  1. A higher “prevalence of comorbid mood disorders” has been mentioned (such as Line 16). Mood disorders and manic-like behaviors should be described in the manuscript in detail.

            Answer: This is a good point. This has been added to the manuscript. This can be found in the ADHD and Bipolar Disorder section.

  1. As discussed in Lines 95-96, the involved brain regions can be summarized for the study among different related diseases.

Answer: more as been added to this area.

Reviewer 2 Report

The manucript deserves publication.

I recommend to add a short paragraph discussing the variation of dopamine receptors in comparison to recent loci identified by GWAS studies for both ADHD and Bipolar Disorders

Demontis, D., Walters, R.K., Martin, J. et al. Discovery of the first genome-wide significant risk loci for attention deficit/hyperactivity disorder. Nat Genet 51, 63–75 (2019). https://doi.org/10.1038/s41588-018-0269-7.

Mullins, N., Forstner, A.J., O’Connell, K.S. et al. Genome-wide association study of more than 40,000 bipolar disorder cases provides new insights into the underlying biology. Nat Genet 53, 817–829 (2021). https://doi.org/10.1038/s41588-021-00857-4.

Author Response

I recommend to add a short paragraph discussing the variation of dopamine receptors in comparison to recent loci identified by GWAS studies for both ADHD and Bipolar Disorders

Answer: This is a great point. Thank you for the references as well. This has been added to the manuscript in the revision.

Reviewer 3 Report

A very good paper. Congratulations to the authors.

Author Response

A very good paper. Congratulations to the authors.

Answer: Thank you for your kind words and for taking the time to read our manuscript. We are very appreciative.